# Measuring the Alignment of ANNs and Primate V1 on Luminance and Contrast Response Characteristics

**Stephanie Olaiya**[1,*]**, Tiago Marques**[2,3,4,5,6,*,§]**, James J. DiCarlo**[2,3,4]

[1]Department of Neuroscience, Brown University
[2]Department of Brain and Cognitive Sciences, MIT, Cambridge, MA02139
[3]McGovern Institute for Brain Research, MIT, Cambridge, MA02139
[4]Center for Brains, Minds and Machines, MIT, Cambridge, MA02139
[6]Champalimaud Clinical Centre, Champalimaud Foundation, Lisbon

[*] Joint first authors (equal contribution)
[§] Corresponding author: tiago.marques@research.fchampalimaud.org

## Abstract

Some artificial neural networks (ANNs) are the current state-of-the-art in modeling the primate ventral stream and object recognition behavior. However, how well they align with luminance and contrast processing in early visual areas is not known. Here, we compared luminance and contrast processing in ANN models of V1 and primate V1 at the level of single-neuron. Model neurons have luminance and contrast response characteristics that differ from those observed in macaque V1 neurons. In particular, model neurons have responses weakly modulated by changes in luminance and show non-saturating responses to high contrast stimuli. While no model perfectly matches macaque V1, there is great variability in their V1-alignment. Variability in luminance and contrast scores is not correlated suggesting that there are trade-offs in the model space of ANN V1 models.

## 1 Introduction

The primate ventral visual stream is a set of hierarchically-organized cortical areas that support visually guided behaviors such as object recognition [1, 2]. Understanding the computations that give rise to this complex visual behavior has been a major goal in visual neurosciences [3]. Over the past years, some artificial neural networks (ANNs), which have also achieved human level visual abilities in computer vision tasks [4], have been used to explain with unparalleled success the response patterns of neurons along the visual ventral stream areas [5, 6, 7, 8]. However, this model to brain comparisons have received some criticism due to the fitting methods that linearly combine thousands of model features to explain the responses of a few biological neurons [9, 10].

Recently, a novel approach for comparing models to brains that does not require the traditional fitting procedure has been proposed [11]. This method explicitly maps single artificial model neurons to single biological neurons and compares the distribution similarity of response properties between the two. Another advantage of this approach is that it can leverage on existing published studies, for which raw data is not accessible, and turn them into quantitative tests for evaluating the model to brain similarity. Using this strategy, Marques et al. 2021, showed that model neurons have surprisingly similar response properties to neurons in primate V1 and that the population distributions in the two systems were closely matched [11]. Despite this, no model in a large pool of candidate models was able to perfectly account for all the responses properties studied.

4th Workshop on Shared Visual Representations in Human and Machine Visual Intelligence (SVRHM) at the Neural Information Processing Systems (NeurIPS) conference 2022. New Orleans.

Here, we extended this approach to study the alignment of ANNs and primate V1 on luminance and contrast response characteristics. There is a vast literature characterizing in detail the responses of neurons in early visual areas to changes in luminance (linear measure of light, spectrally weighted for normal vision, and measured in cd/m$^2$) and contrast (ratio of intensity between the brightest white and the darkest black of a stimulus). These properties are particularly interesting since most ANNs struggle to generalize to distributional shifts in both the brightness and contrast of inputs [12]. It remained to be seen whether the failure of models to deal with brightness and contrast perturbations was due to their lack of alignment with primate V1 luminance and contrast processing at the single neuron level. We make the following novel contributions:

1. We developed 8 new benchmarks that measure the alignment of models to primate V1 on luminance and contrast response characteristics.

2. Using these benchmarks we evaluated a pool of 15 candidate models of primate V1 from pre-assigned layers of ANN models. Luminance scores range from 0 to 0.79 and contrast scores range from 0.51 to 0.76, thus showing that no model studied is able to emulate primate V1 contrast and luminance processing.

3. We compared the luminance and contrast scores with the other V1 response properties scores and observed that these are on average weakly correlated, suggesting that the space of V1 models originating from ANN layers has trade-offs in their alignment to V1.

## 2 Methods

To measure the alignment between the ANNs luminance and contrast processing and that of primate V1, we adapted an existing methodology [11]. In summary, it consists of extracting primate V1 data related to luminance and contrast response properties from existing studies and replicating those experiments in-silico in an ANN model of V1. Then, the distributions of those response properties in both primate V1 and the model are quantitatively compared using a similarity metric. We used data from Kinoshita and Komatsu 2001 for the luminance benchmarks [13] and Sclar et al. 1990 for the contrast benchmarks [14].

### 2.1 Visual Stimuli

We designed visual stimuli to approximate those used in the two studies that provided the primate V1 data. Since models are trained using images with RGB encoded pixel values, when implementing our stimuli, we used a gamma compression function that maps from the physical luminance space to the digital image RGB encoded space (we used the standard range of 0 to 120 cd/m$^2$ to correspond to the whole dynamic range of the input RGB values).

For the luminance response properties, we used seven gray uniform luminance squares which subtended 10 degrees of visual space [13]. The luminance of the squares varied from 0.1 to 100 cd/m$^2$ on a logarithmic scale. For the contrast response properties, we used square achromatic sinusoidal gratings [14]. To determine the preferred spatial frequency and orientation of each neuron, we presented gratings of 8 orientations (spaced by 22.5 deg) and 4 spatial frequencies (from 0.77 to 6.17 cpd on a log scale). For each neuron, we selected the spatial frequency and orientation combination that elicited the strongest response. The gratings were 2.8 by 3.4 deg and were displayed on a black background. To measure neural responses to contrast, the contrast of the gratings varied from 0.02 to 1 logarithmically, with the maximum contrast corresponding to 120 cd/m2.

### 2.2 Recording from ANN-based V1 models

For each ANN, a layer was pre-committed to V1 using a publicly available neural predicitivity benchmark from the Brain-Score platform [8]. Only neurons from the pre-committed V1 layer were analyzed in this study. Stimuli were centered at a specific location near the fovea (0.5 deg eccentricity on both axes). We measured the functional receptive fields (RFs) of all V1 model neurons using a grid of small gratings as previously described [11]. Only neurons that had their RF center at the location around the stimulus center were considered for further analysis. Model activations were transformed to neuronal firing rates using an affine transformation as in [11].

We analyzed the responses of the neurons similarly to corresponding studies. For the luminance experiment, neuronal activations were recorded without baseline correction. For the contrast experiment, we considered either the response's first harmonic of the Fourier transform (AC component) or the mean response with baseline correction (DC component) depending on whether the neuron was a simple or complex cell.

## 2.3 Calculating response properties

We calculated four luminance response properties from the luminance tuning curves: surface responsive, bright slope, dark slope, and normalized delta slope. Surface responsive classifies neurons as surface responsive or non-responsive. Kinoshita et al classified a neuron as surface responsive if its response to the brightest or darkest stimuli was significantly different from its response to the gray ($3cd/m^2$) stimulus [13]. Only surface responsive neurons were used in creating the distributions for the other three response properties. The bright slope property was the slope of the regression line when considering responses to the 'bright stimuli' (luminance of $3cd/m^2$ and higher). The dark slope was the slope of the regression line when considering responses to the dark stimuli ($3cd/m^2$ and lower). The normalized delta slope quantifies the normalized difference between the bright slope and the dark slope.

We calculated four contrast response properties from the contrast tuning curves of each ANN neuron: standard neuron, maximum response, exponent, and semi-saturation constant. Firstly the responses of

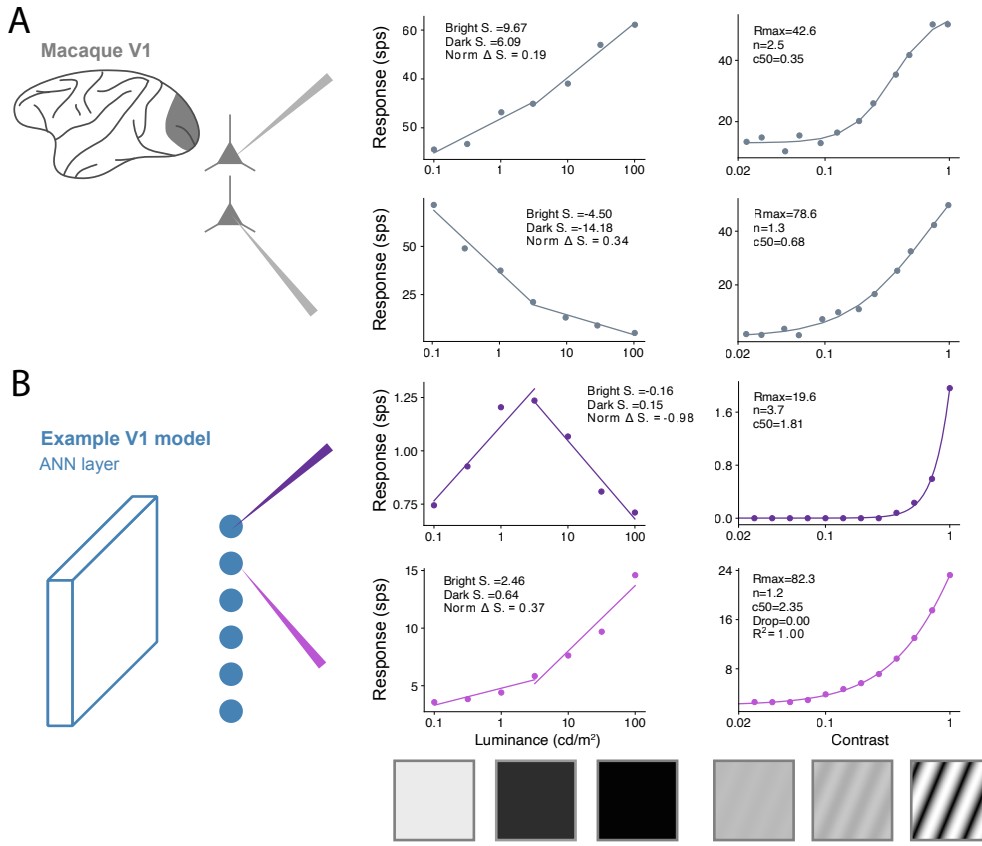

Figure 1: **Measuring luminance and contrast response properties in single neurons of ANN models of V1. A.** Example neuronal responses in macaque V1 [13, 14]. Left column, luminance tuning curve; right curve, contrast tuning curve. Responses are represented in circles and fits to responses with gray curves. Response properties for example neurons are shown on top of each plot. **B.** Same as in **A** but for two example neurons from the same V1 model. Each row corresponds an example neuron. Examples of stimuli used in the experiment are shown at the bottom.

each neuron are fitted using the hyperbolic ratio function which is used in Sclar et al. to characterize contrast response relationships [15, 14]:

$$R = R_{max}\frac{c^n}{c^n+c_{50}^n} \tag{1}$$

The standard neuron response property was a classification that describes whether an artificial neuron had standard V1 neuron responses to contrast. We characterized this as having a peak drop at the highest contrast < 20% and having a good fit to the hyperbolic equation (an $R^2 > 0.9$). Only neurons that were classified as standard were included in the distributions for the other response properties. From the fitted parameters, we extracted the other three properties as maximum response (Rmax), exponent (n), and semi-saturation constant ($C_{50}$).

### 2.4 Comparing response property distributions:

Empirical data for the neuronal response property distributions were extracted from the two studies using the WebPlotDigitizer. We replicated both the luminance and the contrast studies 1000 times by performing in-silico neurophysiological experiments in the models. For each experiment, we randomly sampled the same number of neurons used in each study and computed the distribution of the several response properties. Primate V1 distributions and ANN V1 distributions were compared using a similarity metric that relies on the Kolmogorov-Smirnov Distance (previously described in [11]). This metric provides a quantitative score of the alignment of the ANN V1 distribution to that of primate V1 as seen in the experimental study.

## 3 Results

We analyzed 15 models in our study: AlexNet, ResNet18, ResNet34, ResNet50, ResNet50AT (adversarially trained), ResNet50SIN-IN-IN (trained with style transfer), CORnet-Z, CORnet-S, VOneResNet-50NS (non-stochastic), vgg-16, vgg-19, densenet-121, densenet-169, pnasnet and nasnet [16, 17, 18, 19, 20, 21, 22, 23]. Most model neurons show varying luminance response profiles (Figure 1). These range from monotonic profiles in which responses increase or decrease with increasing luminance to V-shaped responses where the neurons are either gray preferring (peak at intermediate luminance values) or have the lowest responses to intermediate luminance values. In terms of contrast response characteristics, most model neurons have contrast tuning curves with neurons showing non-saturating responses to high contrast.

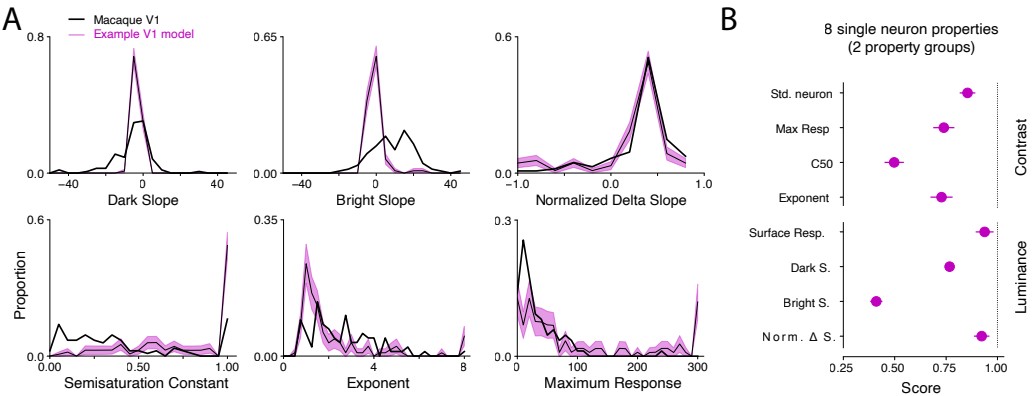

Figure 2: **Distributions of luminance and contrast response properties in a candidate V1 model and macaque V1. A.** Distributions of 6 example response properties in macaque V1 (from published studies, black line) and the best V1 model in the pool (ResNet50AT, purple). Model distributions are obtained by performing in silico experiments, thick line is mean over 1,000 experiments and the shaded region is the SD. **B.** Similarity scores for the eight single neuron response properties for the same V1 model (error bars represent mean and SD). Response properties are grouped in the two categories: luminance and contrast.

Comparing the distributions of response properties in the models with those in macaque V1, we observe that these show striking differences (Figure 2 shows the distributions and corresponding scores for the best performing model). In particular, the response properties with the lowest scores are the semi-saturation constant and bright slope. Analyzing the tuning curves of individual neurons provides some clues for these differences. Unlike primate V1 neurons, model V1 neurons are weakly modulated by luminance. This is seen as most neurons do not have large changes in responses to changes in surface luminance (Figures 1 and 2). Thus, their bright slopes are not large in magnitude. Also, unlike primate V1 neurons, most ANN V1 neurons do not show saturating responses for high contrast stimuli, resulting in much larger $C_{50}$ and Rmax values than those in primate V1 (Figures 1 and 2). Our analysis also showed that most models have a lower proportion of standard and surface responsive neurons than are seen in primate V1. Finally, across all models analyzed, Resnet50AT is the highest performing model on average for both luminance and contrast benchmarks (Figure 2).

Despite all models failing to match primate V1 in all the response properties, how well they align to V1 varies considerably. Luminance scores (average of the corresponding four properties) range from 0 to 0.79 while contrast scores (average of the corresponding four properties) range from 0.51 to 0.76. Interestingly, there is no correlation between luminance and contrast scores in the models studied (Figure 3A). When comparing the scores on these benchmarks with those on the other seven V1 response property benchmark categories (corresponding to 22 individual benchmarks [11]), we observe a similar trend: while some pairs of benchmarks appear to be correlated, on average, pair-wise correlations between scores on V1 benchmarks are very weak. This suggests that there are trade-offs in this model space in terms of alignment to multiple aspects of V1 processing (Figure 3B).

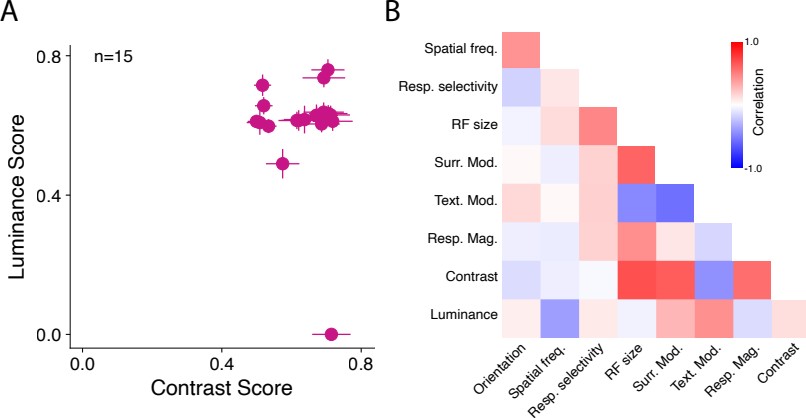

Figure 3: **Luminance and contrast scores are not correlated with each other and with other V1 response property benchmarks. A.** Luminance and contrast average scores for the 15 models analyzed in this study (each score is the average of the four corresponding response property benchmarks). **B.** Pair-wise correlations between the nine V1 response property categories scores. Each category is the average of multiple individual benchmarks [11]

## 4  Discussion

In this work, we evaluated whether ANN models of V1 are aligned to macaque V1 luminance and contrast processing at the single-neuron level. While most model neurons respond to contrast and luminance stimuli, their responses differ from those observed in populations of macaque V1 neurons. In particular, model neurons have responses weakly modulated by changes in luminance and show non-saturating responses to high contrast stimuli. Furthermore, models differ in their alignment to V1 with the best model being one adversarially-trained, a result that is consistent with previous findings for other benchmarks [22, 11].

While we view this study as an important first step to studying luminance and contrast processing in ANN models of V1, much work remains to be done. We sampled a considerable pool of models with varying architectures and training procedures. However, the space of V1 models is considerably larger and future work should explore more models with more diverse architectures, as well as analyzing

the alignment to V1 of other layers in the same hierarchical models (no layer pre-commitment). Furthermore, comparisons of luminance and contrast scores with other brain-benchmarks, particularly those in higher visual areas and behavior, may provide insight into how luminance and contrast processing in early vision relates to downstream processing.

## Acknowledgments and Disclosure of Funding

This work was started during the MIT Summer Research Program (MSRP). S.O. would like to thank the director of the program Mandana Sassanfar and MIT for this research opportunity. S.O. would also like to thank the members of the DiCarlo lab and the other MSRP students for the critical discussions and feedback. This work was supported by the PhRMA Foundation Postdoctoral Fellowship in Informatics (T.M.), the Semiconductor Research Corporation (SRC) and DARPA (J.J.D.), Office of Naval Research grant MURI-114407 (J.J.D.), the Simons Foundation grant SCGB-542965 (J.J.D.), the MIT-IBM Watson AI Lab grant W1771646 (J.J.D.).

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
