# OpenReview forum: "Measuring the Alignment of ANNs and Primate V1 on Luminance and Contrast Response Characteristics"
_NeurIPS.cc/2022/Workshop/SVRHM — SVRHM Poster_

### Official Review · Reviewer_CJGk · 2022-10-13
**Some interesting findings and a new tool**

**Rating:** 7
**Confidence:** 4

**Review:**

This paper looks at the luminance and contrast responses of neurons in pre-trained artificial models and compares them to neurophysiological recordings from previous studies of response rate in a Macaque V1. The overall finding is that there are some significant differences in behaviour of the artificial models to the neurophysiological data.

Overall, I think there are a number of interesting findings from this work that are of value for the SRVHM audience. From a model engineering perspective this paper gives a way forward to looking at what features of ANN design might make the luminance and contrast responses more like the cellular recordings. I have a couple of questions that came to mind when reading:

 - I understand that the stimuli were gamma corrected, but presumably the data the networks were trained on was not? Is this important?
 - The approach looks at V1 activity, and essentially ignores the retina/eye in both cases. Could this be important though? The primate retina & eye have a number of dynamic features that (probably) help it adjust its response to its input; it's not clear that he ANN models really have the same features.

I'd like to ask that the authors clarify fig 3 as this is one of the most interesting parts of the paper. 3A would benefit from indicating the actual models (at least for the extremal ones). In Fig 3B can you clarify the labels (I can guess without reading [11] but it would be nice not to).

---

### Official Review · Reviewer_jrji · 2022-10-13
**Nice first steps of an interesting line of work**

**Rating:** 8
**Confidence:** 3

**Review:**

In this work, the authors perform in-silico experiments on neurons of a set of DNNs, analogous to experiments performed in real V1 neurons, and compare the response properties of these two types of neurons. In particular, they analyze two very basic response properties of neurons: response to luminance and to contrast changes.

The methods seem sound for approaching the study’s aim, and are for the most part clearly explained. Particularly, I like that the authors pre-selected the layers to be compared to V1 by their Brain-Score performance.

What I find most interesting about the results and aim of this work is that they analyze some very basic response properties that one thinks should be among the easiest to interpret and understand in the future. It is interesting to wonder whether DNN neurons may be better at explaining the responses of V1 neurons to more complex stimuli than to the extremely simple stimuli presented here (although this is not something discussed in the paper, I think it’s an interesting question that this work may help advance).

From what I see, I don’t have any concerns or really any major comments to make on the quality work. The methods seem sound, the results are interesting, and both are clearly and soberly communicated.

One comment I would make is that I don’t find it clear in the manuscript what the authors expect to contribute to the literature from this analysis and whatever work follows up. As they state it, I understand that what they do is an extension of previous work, where they “simply” add real/model neuron comparisons on a couple of new response properties to a repertoire of existing comparisons. Why would adding these new properties be interesting, or where could this work go? I mentioned one possible interest above (that these should be some of the most interpretable properties), and the authors will probably have other ones, but I don’t think this is clear from the text. There is some attempt at this in lines 34-36, asking whether the failure of DNNs to be robust to brightness and contrast perturbations “is due” to lack of alignment with V1. Although I’m not fully on board with the use of language of saying that such lack of robustness “is due” to lack of alignment with V1, this seems like some good motivation for this work, and the rationale could be better developed. After these couple of sentences, this motivation is abandoned for the rest of the text.

Related to previous points, I wonder whether Brain-Score may put more weight on other response characteristics of V1 neurons, and whether the best Brain-Score layer may not be the best layer for describing this particular property. Doing experiments on this is out of the scope of this work (and the authors point something similar as future work), but maybe they can discuss a bit this point of the possible mismatch between Brain-Score and the response properties that they are analyzing here.

Paragraph starting on line 77: What is a “baseline correction” here? I don’t quite get the AC/DC component thing and dependence on simple/complex. It doesn’t seem essential to understand, but if authors can provide some intuition there, that would be good.

---

### Official Review · Reviewer_6bUZ · 2022-10-14
**Moving towards psychological plausiblity in addition to neural plausibility for CNNs**

**Rating:** 6
**Confidence:** 4

**Review:**

This paper attempts to make CNNs more psychologically plausible by investigating behaviour in addition to neural plausibility. Comparisons between single-neuron recordings of Non-human primate V1 spiking data and Deep Neural Networks neurons are made. Specifically, luminance and contrast response characteristics are investigated & it is reported that DNN neurons have responses weakly modulated by luminance and non-saturating responses to contrast. This is an essential step towards understanding if CNNs are a plausible model for primate ventral visual stream & how they can be improved.

* Strengths:
  1. Comparisons between DNNs & single neuron recordings.
  2. Good stimuli design
  3. Extensive model benchmarking

* Weaknesses:
  * Major:
    1. Since the paper only tests CNN models, it would be a better claim if it was about CNNs rather than ANNs (in the title, abstract, etc.)
    2. The NHP V1 data papers seem fairly old; it would be good to justify the reason behind selecting the data from these papers rather than newer/other papers. Is it due to availability?
    3. While mentioning the selection of neurons for analysis, for example, surface-responsive neurons, it would be helpful to show the number/percentage of neurons analysed instead of stating most neurons. Specifically lines 87, 107, 116, 120, 127, 129 and 131.
    4. Exact model scores are not mentioned; they should be mentioned to have a better understanding of the results.
    5. It would be interesting and helpful to see between model comparisons to demonstrate which property is correlated with scores. For example, scale (number of parameters), brain score, robustness, performance, etc.
  * Minor:
    1. Figure 1 title A, "right curve, contrast tuning curve" -> "right column, contrast tuning curve"
    2. Line 124, mention the best performing model, i.e. ResNet50AT.
    3. In Figure 2A, the first row, please mention the y-label
    4. In Figure 3B, please mention "orientation" in the first row and "Luminance" in the last column.

* Conclusion:
  The paper attempts an important step towards psychological plausibility, but the reporting of the results should be improved to paint a better picture.